# MITIGATING BIAS IN DATASET DISTILLATION

## ABSTRACT

Dataset distillation (DD) has emerged as a technique for compressing large datasets into smaller synthetic counterparts, facilitating downstream training tasks. In this paper, we study the impact of bias within the original dataset on the performance of dataset distillation. With a comprehensive empirical evaluation on datasets with color, corruption and background biases, we found that color and background biases in the original dataset will be amplified through the distillation process, resulting in a notable decline in the performance of models trained on the synthetic set, while corruption bias is suppressed through the distillation process. To reduce bias amplification in dataset distillation, we introduce a simple yet highly effective approach based on a sample reweighting scheme utilizing kernel density estimation. Empirical results on multiple datasets demonstrated the effectiveness of the proposed method. Notably, on CMNIST with 5% bias-conflict ratio and IPC 50, our method achieves 91.5% test accuracy compared to 23.8% from vanilla DM, boosting the performance by 67.7%, whereas applying state-of-the-art debiasing method on the same synthetic set only achieves 53.7%. Our findings highlight the importance of addressing biases in dataset distillation and provide a promising avenue to mitigate bias amplification in the process.

## 1 INTRODUCTION

Dataset plays a central role in the performance of machine learning models. With advanced data collection and labeling tools, it becomes easier than ever to construct large-scale datasets. The rapidly growing size of contemporary datasets not only poses challenges to data storage and preprocessing, but also makes it increasingly expensive to train machine learning models and design new methods, such as architectures, hyperparameters, and loss functions. As a result, dataset distillation (Wang et al., 2018) (also known as dataset condensation) emerges as a promising direction for solving this issue. Dataset Distillation aims at compressing the original large-scale dataset into a small subset of information-rich examples, enabling models trained on them to achieve competitive performance compared with training on the whole dataset. The distilled dataset with a significantly reduced size can therefore be used to accelerate model training and reduce data storage. Dataset distillation has been shown to benefit a wide range of machine learning tasks, such as Neural Architecture Search (Wang et al., 2021), Federated Learning (Xiong et al., 2023), Continual Learning (Wiewel & Yang, 2021), Graph Compression (Jin et al., 2021), and MultiModal training (Wu et al., 2023).

Recent dataset distillation methods (Loo et al., 2023; Zhao et al., 2023; Du et al., 2023; Cui et al., 2023) mainly target performance enhancement on standard datasets like MNIST (Deng, 2012) and CIFAR-10 (Krizhevsky et al., 2009). Despite their focus, dataset bias (Tommasi et al., 2017), a significant issue in machine learning, is often overlooked in the context of dataset distillation. Dataset bias emerges when collected data unintentionally reflects existing biases, leading to skewed predictions and potential ethical concerns. Although bias detection and mitigation strategies have been extensively studied in the past few years (Sagawa et al., 2019; Li & Vasconcelos, 2019; Nam et al., 2020; Lee et al., 2021; Hwang et al., 2022), the impact of dataset bias on data distillation remains unexplored. Since a biased synthetic set can result in inaccurate or unfair decisions, it is important to understand the role of bias in dataset distillation and how to mitigate bias in such processes.

This paper provides a novel study of how biases in the original training set affect dataset distillation process. Specifically, we are interested in the following questions: *1). How does bias propagate from the original dataset to the distilled dataset? 2). How do we mitigate biases present in the distilled dataset?*

To answer the first question, we assess existing dataset distillation algorithms across several benchmark datasets. Our findings reveal that the distillation process is significantly influenced by the type of bias, with color and background biases being amplified and noise bias suppressed. For datasets exhibiting amplified biases, we found even state-of-the-art de-biasing training methods are insufficient in restoring the original performance. This highlights the urgency of developing a bias-mitigating dataset distillation algorithm.

To counter bias in distilled datasets, we propose a simple yet effective debiasing algorithm based on data point reweighting. Leveraging the insight that biased data points cluster in the model's embedding space, we down-weight samples within such clusters using kernel density estimation. This re-weighting rebalances the significance of biased and unbiased samples, mitigating biases in the distillation process. Empirical results on diverse bias-injected datasets demonstrate that the proposed reweighting scheme significantly reduces bias in the distilled datasets. For example, on Colored MNIST with a 5% bias in conflicting samples and 50 images per class, the original Distribution Matching (DM) method leads to a biased synthetic set. A model trained on such a synthetic set achieves only 23.8% accuracy. In contrast, our reweighting method produces a more unbiased dataset, resulting in 91.5% accuracy, representing a 67.7% performance improvement over DM.

In summary, our contributions are: 1) We provide the first study on the impact of biases in dataset distillation process. 2) We propose a simple yet effective re-weighting scheme to mitigate biases in two types of DD methods. 3) Through extensive experiments and ablation studies, we demonstrate the effectiveness of our methods.

## 2 RELATED WORK

### 2.1 DATASET DISTILLATION

Dataset Distillation (DD) aims to compress a large dataset into a small but informative one to achieve competitive performances compared to the whole dataset. Due to the greatly reduced dataset size and competitive performance, it can also be used for many downstream tasks as mentioned in Section 1. There are mainly two lines of work in dataset distillation including ones that directly operate at pixel level (non-parameterization) (Wang et al., 2018; Zhao & Bilen, 2021a; Wang et al., 2022; Zhou et al., 2022; Loo et al., 2022; Cazenavette et al., 2022; Liu et al., 2023) and the other ones that utilize parameterization to compress images into more compact forms (Kim et al., 2022; Deng & Russakovsky, 2022; Liu et al., 2022; Lee et al., 2022; Cazenavette et al., 2023). Since most of the parameterization methods can be used as an add-on module to non-parameterization methods, we focus on non-parameterization methods to study the effect of bias in this paper. Among the recent methods, we study 3 methods specifically including DSA, DM and MTT which are the basis for many following state-of-the-art (SOTA) methods. Dataset Condensation with Differentiable Siamese Augmentation (DSA) (Zhao & Bilen, 2021b) is a bi-level optimization process to distill dataset by matching the model gradients generated by real data and the distilled dataset. Dataset Condensation With Distribution Matching (DM) (Zhao & Bilen, 2021a) identifies the huge computation cost in bi-level optimizations and generates synthetic datasets by directly matching the distribution of real data and distilled dataset in the embedding space. Dataset Distillation by Matching Training Trajectories (MTT) (Cazenavette et al., 2022) tries to match the training trajectories between models trained using synthetic dataset and models trained using the whole training dataset.

### 2.2 DATASET AND MODEL BIAS

Deep neural networks (DNNs) have exhibited remarkable capacity in discovering strong correlations present within datasets, which often contributes to their success in various machine learning tasks. However, when applied to datasets where simple and spurious correlations coexist with complex and intrinsic correlations, DNNs may inadvertently lean towards the shortcuts. The generalization ability of DNNs will be greatly hindered when these spurious relations are learned instead of the intrinsic ones. Following previous works (Hwang et al., 2022), we refer to samples strongly correlates with bias feature as bias-aligned and samples that don't align with bias features as bias-conflicting. The most commonly studied bias types include color, background, noise, texture etc (Nam et al., 2020; Lee et al., 2021; Hwang et al., 2022). In this paper, we study how the bias in the original data affects the small synthetic set through the dataset distillation process and try to mitigate this phenomena.

Figure 1: Examples of original images (top) and synthetic images (bottom) from biased dataset. See Section A.9 for more examples.

The evaluations are done by testing whether models trained on synthetic datasets can effectively generalize to unbiased test datasets.

### 2.3 DE-BIASING METHODS

While prior research has explored de-biasing methods on entire datasets, our work is the first, to the best of our knowledge, to investigate the impact of biases on dataset distillation results in this emerging field. Below, we discuss several state-of-the-art de-biasing methods designed to train de-biased models on entire datasets.

LfF (Nam et al., 2020) debiases by training a biased and debiased model together. It uses generalized cross entropy (GCE) (Zhang & Sabuncu, 2018) loss to amplify bias and computes a difficulty score for debiased model loss weighting. DFA (Lee et al., 2021) disentangles bias and intrinsic attributes, creating unbiased samples by swapping bias embeddings among training samples to train a debiased model. SelectMix (Hwang et al., 2022) shares the same idea of creating more unbiased samples by using mixup augmentation on contradicting pairs selected by a biased model.

Note that, although some prior de-biasing methods involve reweighting (Sagawa et al., 2019; Li & Vasconcelos, 2019; Nam et al., 2020), our approach differs for the following reasons: 1) Previous methods (Li & Vasconcelos, 2019; Nam et al., 2020) integrate the reweighting scheme into the final de-biased model's training, either as a loss or using an auxiliary model, optimizing it alongside the de-biased model. This process does not apply to dataset distillation methods since a well-trained model is not required in the process (e.g., DM uses randomly initialized models, and MTT only matches part of the training trajectories). 2) Methods like Sagawa et al. (2019) require explicit bias supervision, which may be challenging or infeasible, while our method has no such requirement.

## 3 THE IMPACT OF BIAS IN DATASET DISTILLATION

In this section, we conduct comprehensive experiments to answer the following question: *how does a biased training set influence the distilled data? Will the bias be amplified or suppressed through the dataset distillation process?*

In line with prior studies (Nam et al., 2020; Lee et al., 2021; Hwang et al., 2022), we explore three datasets: Colored MNIST (CMNIST), Background Fashion-MNIST (BG FMNIST), and Corrupted CIFAR-10. CMNIST introduces a color bias, causing classes to share specific colors, potentially confusing training. BG FMNIST combines MiniPlaces (Zhou et al., 2017) with Fashion-MNIST, creating background biases, e.g. T-shirts in bamboo forests. Corrupted CIFAR-10 introduces perturbations and image distortions like Gaussian noise, blur, brightness, contrast changes, and occlusions. See Figure 1 for examples and Section 5.1 for detailed descriptions.

For each dataset, we use $D$ to denote the unbiased set (e.g., randomly distributed colors in CMNIST), while $D_b$ represents the bias-injected dataset. In $D_b$, 95% of the samples are aligned with the bias, meaning, for example, 95% of the digit '0' may be red, while the remaining 5% of the digit '0' possess random colors. Let $\mathcal{F}$ be a dataset distillation algorithm that maps the original dataset into a distilled synthetic set, and let $\mathcal{M}$ denote the model training procedure that maps a dataset to a model. By comparing $\mathcal{M}(\mathcal{F}(D))$ (model trained with distilled unbiased dataset) and $\mathcal{M}(\mathcal{F}(D_b))$ (model trained with distilled biased dataset), we evaluate these two models' performance on unbiased test samples and compute the following measurement to reveal how the bias in the source dataset degrades the performance of the model trained on distilled synthetic samples:

$$\text{Acc}(\mathcal{M}(\mathcal{F}(D))) - \text{Acc}(\mathcal{M}(\mathcal{F}(D_b))).$$

Figure 2: The left most 2 bars indicate the model performance on full dataset with no distillation. For DSA/DM/MTT, the blue bar shows the model performance on the unbiased dataset and the red bar shows the performance of the corresponding dataset distillation method on that biased dataset with 5% bias-conflicting samples. The distillation performances are measured under IPC 10.

We conduct experiments on three representitive dataset distillation algorithms: DSA (Zhao & Bilen, 2021b), DM (Zhao & Bilen, 2021a) and MTT (Cazenavette et al., 2022) as lots of SOTA methods can be used as an add-on module to these methods (Wang et al., 2022; Kim et al., 2022; Liu et al., 2022; Lee et al., 2022; Liu et al., 2023). Additionally, we include a baseline approach without dataset distillation, which is equivalent to the case when $\mathcal{F}$ represents an identity transformation. The visualization of bias impacts in dataset distillation can be found in Figure 2.

For CMNIST, the results presented on the left panel of Figure 2 reveal a strong **bias amplification** effect – while the bias injected into the original dataset leads to a mere 4% performance drop in regular training, it results in over 50% performance decline when employing any of the three dataset distillation methods. This observation can be attributed to the following factors. Since the original set comprises 95% biased samples, with a selection of IPC 10, it is highly possible that all of the chosen images are biased. As a result, the unbiased signal totally diminishes through the distillation process. Another critical factor is that since color is a discriminative feature that can be easily learned by neural networks, dataset distillation algorithms will focus on distilling this color feature into the synthetic images in order to achieve good performance, leading to bias amplification. Similar impacts are also seen on the BG FMNIST dataset.

Interestingly, results from Corrupted CIFAR-10 show a reverse trend. The right panel of Figure 2 shows that the performance degradation is actually more substantial in the traditional training pipeline compared to the ones incorporating dataset distillation. This observation indicates the **bias suppression** effect for corruption biases, and dataset distillation is helpful for mitigating bias in this setting. Since corruption biases include several different perturbation effects such as Gaussian noise and blurring, we assume that the distillation process naturally blends information from multiple images, and the resulting images are already blurred in nature where noisy effects tend to cancel out, so it is harder to capture the corruption bias. Sampled distilled images are visualized in Figure 1.

For datasets exhibiting amplified biases (CMNIST and BG FMNIST), we found even state-of-the-art de-biasing training methods such as SelectMix (Hwang et al., 2022) and DFA (Lee et al., 2021) are not able to obtain an unbiased model from the biased synthetic set. See more details in the first subsection of Section 6. This finding also highlights the urgency of developing a bias-mitigating DD algorithm to obtain unbiased synthetic sets.

## 4 MITIGATING BIAS IN DATASET DISTILLATION

In this section, we propose a simple yet effective re-weighting method to mitigate bias in several dataset distillation algorithms.

### 4.1 BIAS MITIGATION THROUGH RE-WEIGHTING

In dataset distillation methods such as DM (Zhao & Bilen, 2021a; Zhao et al., 2023), the objective is to match the embeddings generated by the synthetic dataset ($\mathcal{S}$) with the ones generated by the real images ($\mathcal{T}$). The objective function is formulated as below

$$\min_{\mathcal{S}} \ \mathbb{E}_{v \sim P_v, \omega \sim \Omega} \parallel \frac{1}{|\mathcal{T}|} \sum_{i=1}^{|\mathcal{T}|} \psi_v(\mathcal{A}(x_i, \omega)) - \frac{1}{|\mathcal{S}|} \sum_{i=1}^{|\mathcal{S}|} \psi_v(\mathcal{A}(s_j, \omega)) \parallel^2, \quad (1)$$

where $\psi$ is usually a surrogate model that maps the data into an embedding space and $\mathcal{A}$ is a differentiable augmentation function. By solving equation 1, DM learns the synthetic set to match the mean embedding of the original set. If the dataset is highly biased, the first term $\frac{1}{|\mathcal{T}|}\sum_{i=1}^{|\mathcal{T}|}\psi_v(\mathcal{A}(x_i,\omega))$ will be dominated by bias-aligned samples, thus causing distribution matching based methods to synthesize more images that's also bias-aligned.

In order to mitigate bias, we propose to compute a weighted sum of the real image embeddings instead of simply using the mean of all data points. For data points that exhibit a strong correlation with the spurious (bias) feature, they should be assigned lower weights. Conversely, data points that have limited association with the bias feature should be afforded higher weights. This adjustment ensures that the distillation process effectively captures the intrinsic features. Let $W(\mathcal{T}) = [w_0, w_1, ..., w_n]$ with $n$ equals $|\mathcal{T}|$ in Equation 1 be the normalized weight of each training sample, the weighted loss can be written as

$$\min_{\mathcal{S}}\ \mathbb{E}_{v\sim P_v,\omega\sim\Omega}\ \|\ W(\mathcal{T})\cdot\psi_v(\mathcal{A}(\mathcal{T},\omega)) - \frac{1}{|\mathcal{S}|}\sum_{i=1}^{|\mathcal{S}|}\psi_v(\mathcal{A}(s_j,\omega))\ \|^2, \tag{2}$$

where $W(\mathcal{T})\cdot\psi_v(\mathcal{A}(\mathcal{T},\omega))$ is the re-weighted embeddings where the bias feature has been balanced.

## 4.2 Bias estimation using Kernel Density Estimation

The key problem of this reweighting scheme is how to compute $W(X)$ which is unknown. In order to compute it, we propose to use Kernel Density Estimation (KDE), which is a non-parametric technique used to estimate the probability density function (PDF) of a random variable based on observed data points. Mathematically, given a set of n data points $x_1, x_2, ..., x_n$, the kernel density estimate $\hat{f}(x)$ at any point x is given by:

$$\hat{f}(x) = \frac{1}{n}\sum_{i=1}^{n} K\left(\|\Phi(x) - \Phi(x_i)\|\right), \tag{3}$$

where $K(\cdot)$ is the kernel function that determines the shape and width of the kernel placed on each data point. The most commonly used kernel function is the Gaussian kernel: $K(u) = \frac{1}{\sigma\sqrt{2\pi}}e^{-\frac{1}{2}\cdot(\frac{u}{\sigma})^2}$. $\Phi$ is a mapping function that maps the data points into an embedding space for distance computation. By summing the contributions of the kernel functions centered at each data point, KDE provides an estimate of the PDF at any given point $x$. The estimate $\hat{f}(x)$ represents the density of the underlying distribution at that point. Since a biased dataset is usually dominated by bias-aligned samples which should be given a lower weight, we propose to use the normalized inverse of the kernel density function $\mathbb{N}(\frac{1}{\hat{f}(x)})$ as the new weights, and $\mathbb{N}$ is a normalization function such as softmax so that $\sum_{i=1}^{|\mathcal{T}|}\mathbb{N}(\frac{1}{\hat{f}(x_i)}) = 1$. Eventually we have $W(\mathcal{T}) = [\mathbb{N}(\frac{1}{\hat{f}(x_1)}), ..\mathbb{N}(\frac{1}{\hat{f}(x_n)})]$.

## 4.3 distance computation in KDE

The aforementioned KDE reweighting scheme requires a feature mapping $\Phi(\cdot)$ to map images from raw pixel space to a more meaningful hidden space. Moreover, an optimal mapping for $\Phi(\cdot)$ would be one that transforms images into biased features, enabling KDE to accurately represent the density of bias. Consequently, our reweighting scheme can effectively mitigate bias. To obtain bias features without external knowledge, Lee et al. (2021); Nam et al. (2020) both utilize the generalized cross entropy (GCE) loss to train an auxiliary bias model. However, neither of them directly work with latent spaces that can be utilized in KDE. Following the recent SOTA de-biasing method (Hwang et al., 2022), we utilize a supervised contrastive learning (Chen et al., 2020; Khosla et al., 2020) model, which is trained with generalized supervised contrastive (GSC) loss to produce image embeddings that can be used to measure distances between data points. The model is proven to produce high quality similarity matrix regarding bias features (Hwang et al., 2022).

### 4.4 APPLY TO OTHER DATASET DISTILLATION METHODS

In addition to DM, our method can also be easily applied to other DD methods. Here we use DSA (Zhao & Bilen, 2021b; 2023; Kim et al., 2022) as an example which synthesizes data by matching gradients. Generally, gradient matching based methods can be formulated as:

$$\min_{\mathcal{S}} \; D(\nabla_\theta \mathcal{L}_c^{\mathcal{T}}(\mathcal{A}(\mathcal{T}, \omega^{\mathcal{T}}), \theta_t), \nabla_\theta \mathcal{L}_c^{\mathcal{S}}(\mathcal{A}(\mathcal{S}, \omega^{\mathcal{S}}), \theta_t)), \tag{4}$$

where $\mathcal{L}_c^{\mathcal{T}} = \frac{1}{|\mathcal{T}|} \sum_{x,y} \ell(\phi_{\theta_t}(\mathcal{A}(\mathcal{T}, \omega^{\mathcal{T}}))$. We denote the re-computed weights as $W(\mathcal{T})$, then the first term in Equation 4 can simply be replaced with $W(\mathcal{T}) \cdot \nabla_\theta \mathcal{L}_c^{\mathcal{T}}(\mathcal{A}(\mathcal{T}, \omega^{\mathcal{T}}), \theta_t)$. We demonstrate that the proposed method works well with both DSA and DM in the following section.

## 5 EXPERIMENTS

### 5.1 DATASETS

We conduct experiments on 3 datasets, two of which are widely used in SOTA de-biasing methods to assess color and noise biases, while also introducing a novel dataset to evaluate background biases.

**Colored MNIST (CMNIST):** Nam et al. (2020) introduces the Colored MNIST dataset by injecting color with random perturbation into the MNIST dataset (Deng, 2012). Each digit will be associated with a specific color as its bias such as digit 0 being red and digit 4 being green. We evaluate our method under 3 bias conflicting ratios with 1% (54,509 bias aligned, 491 bias conflicting), 2% (54,014 bias aligned, 986 bias conflicting) and 5% (52,551 bias aligned, 2,449 bias conflicting)[1].

**Corrupted CIFAR-10:** Generated from the regular CIFAR-10 dataset (Krizhevsky et al., 2009), Corrupted CIFAR-10 applies different corruptions (Hendrycks & Dietterich, 2019) to the images in CIFAR-10 so that images from one class are associated with one type of corruption such as GaussianNoise or MotionBlur. We also evaluate our method under 3 bias conflicting ratios with 1% (44,527 bias aligned, 442 bias conflicting), 2% (44,145 bias aligned, 887 bias conflicting) and 5% (42,820 bias aligned, 2,242 bias conflicting).

**Background Fashion-MNIST (BG FMNIST):** Background bias, which results in an over-reliance on the background for predicting foreground objects, has been employed to assess interpretability methods in various prior studies Yang & Kim (2019); Zhou et al. (2017). Following this idea, we construct a new dataset biased in backgrounds by using Fashion-MNIST (Xiao et al., 2017) as foregrounds which include a training set of 60,000 examples and a test set of 10,000 examples. And MiniPlaces (Zhou et al., 2017) is used as backgrounds to introduce background biases such as T-shirt is associated with bamboo forest background, trouser is associated with livingroom background, etc. Similar to other biased datasets, we also conduct experiments in 3 settings including 1%, 2% and 5% bias conflicting samples.

### 5.2 EXPERIMENTAL SETUP

Following previous dataset distillation methods (Zhao & Bilen, 2021a; Cazenavette et al., 2022), we use ConvNet as the model architecture. It has 128 filters with kernel size of 3×3. Then it's followed by instance normalization, RELU activation, and an average pooling layer. We use SGD as the optimizer with 0.01 learning rate. For the supervised contrastive model, we use ResNet18 (He et al., 2016) following Hwang et al. (2022) with a projection head of 128 dimensions. Same as previous de-biasing works (Nam et al., 2020; Lee et al., 2021; Hwang et al., 2022), we evaluate our results by training a DNN model (same as distillation) on the synthetic dataset and measure its accuracy using the unbiased test set. The results are evaluated with IPC 1, 10 and 50 on CMNIST, Corrupted CIFAR-10 and BG FMNIST dataset. See Section A.2 in Appendix for more experiment details.

---

[1]Bias-conflicting samples are samples in a class that have different bias properties from the majority of the samples in that class. For example, a yellow 0 is a bias conflicting sample when the majority of 0s are red.

Table 1: Test accuracy of distribution matching based method on three biased datasets.

| Dataset | Method | Bias-conflict Ratio (1.0%) | | | Bias-conflict Ratio (2.0%) | | | Bias-conflict Ratio (5.0%) | | |
|---|---|---|---|---|---|---|---|---|---|---|
| | | 1 | 10 | 50 | 1 | 10 | 50 | 1 | 10 | 50 |
| CMNIST | DM | 25.4±0.1 | 18.6±0.2 | 22.6±0.5 | 24.8±0.4 | 18.5±0.6 | 23.6±0.8 | 25.3±0.3 | 19.6±0.9 | 23.8±1.3 |
| | DM+Ours | **28.0±0.5** | **64.9±0.3** | **75.4±1.1** | **26.4±0.7** | **50.6±1.2** | **75.7±1.0** | **32.2±1.0** | **86.5±1.2** | **91.5±0.9** |
| BG FMNIST | DM | 41.0±0.3 | 42.2±0.8 | 43.9±0.4 | 40.1±0.6 | 40.1±0.9 | 44.4±0.5 | 41.7±0.5 | 42.0±1.2 | 44.6±0.9 |
| | DM+Ours | **44.6±0.5** | **50.6±0.2** | **57.2±0.6** | **51.4±0.7** | **62.3±0.4** | **63.0±1.0** | **49.4±0.2** | **61.8±0.6** | **65.0±0.8** |
| Corrupted CIFAR-10 | DM | **25.1±0.4** | 32.9±0.3 | 37.6±0.8 | 25.0±0.1 | 32.9±0.1 | 37.7±0.2 | 24.6±0.4 | **33.5±0.8** | 38.7±0.4 |
| | DM+Ours | 24.2±1.2 | **33.4±0.9** | **39.4±0.8** | **25.3±0.5** | **34.2±0.5** | **39.7±0.4** | **26.6±0.5** | 33.5±0.6 | **40.2±0.4** |

## 5.3 EXPERIMENTAL RESULTS

### 5.3.1 PERFORMANCE BOOST ON DM

First of all, we investigate if the proposed method can mitigate biases in distribution matching based methods and choose DM (Zhao & Bilen, 2021a) as the representative algorithm. The evaluation results are shown in table 1. It can be seen from the table that as the number of IPCs increases, the vanilla DM has little to no performance gain on bias amplifying datasets CMNIST and BG FMNIST. When IPC increases from 1 to 10 on CMNIST with 1, 2 and 5 percent bias-conflicting samples, there is even a performance degradation. After reweighting the samples according to Equation 2, we are able to mitigate the biases in synthetic datasets. Under the settings of IPC 50, we are able to boost the performance from 22.6% to 75.4% on CMNIST dataset with 1% bias conflicting samples. With 2% and 5% bias conflicting samples, the accuracy also increased from 23.6% to 75.7% and 23.8% to 91.5% respectively. The complete results can be found in table 1. Similar performance boosts are also observed on BG FMNIST, e.g. with IPC 10, the performance gains are 8.4%, 22.2% and 19.8% with 1, 2 and 5 percent bias conflicting samples. On Corrupted CIFAR-10, we only observe a slight performance boost which aligns with the intuition described in Section 3.

### 5.3.2 PERFORMANCE BOOST ON DSA

Next, we investigate whether the proposed method can improve gradient matching based methods and choose DSA as the representative algorithm. Similar to DM, we also observe a strong performance boost on CMNIST. Under the settings of IPC 50, with 1, 2 and 5 percent bias conflicting samples, the performance increases from 14.5% to 81.4%, 30.9% to 83.0% and 68.5% to 94.0%. On BG FMNIST, the performances increase from 40.7% to 58.3%, 48.4% to 65.1% and 59.3% to 71.2% for 1, 2 and 5% bias conflicting samples with IPC 50. Complete results are shown in table 2.

Table 2: Test accuracy of gradient matching based method on three biased datasets.

| Dataset | Method | Bias-conflict Ratio (1.0%) | | | Bias-conflict Ratio (2.0%) | | | Bias-conflict Ratio (5.0%) | | |
|---|---|---|---|---|---|---|---|---|---|---|
| | | 1 | 10 | 50 | 1 | 10 | 50 | 1 | 10 | 50 |
| CMNIST | DSA | 26.1±0.3 | 16.5±0.2 | 14.5±0.2 | 25.2±0.3 | 16.8±0.3 | 30.9±0.4 | 25.9±0.5 | 27.3±0.4 | 68.5±1.2 |
| | DSA+Ours | **27.9±0.4** | **76.7±1.1** | **81.4±0.8** | **26.4±0.2** | **75.3±0.3** | **83.0±1.2** | **32.6±0.1** | **91.9±0.7** | **94.0±0.8** |
| BG FMNIST | DSA | 43.4±0.4 | 45.8±0.5 | 40.7±0.9 | 43.7±0.5 | 47.6±0.3 | 48.4±0.8 | 44.7±0.6 | 52.8±0.5 | 59.3±0.6 |
| | DSA+Ours | **44.4±0.6** | **57.0±1.0** | **58.3±0.8** | **48.5±1.2** | **64.4±0.9** | **65.1±0.8** | **46.2±0.6** | **66.4±0.6** | **71.2±1.1** |
| Corrupted CIFAR-10 | DSA | 25.5±0.3 | 31.9±0.8 | 34.1±0.5 | 25.1±0.2 | 32.0±0.1 | 34.2±0.3 | 25.7±0.5 | **32.8±0.6** | 35.6±0.5 |
| | DSA+Ours | **26.0±0.1** | **32.6±0.8** | **35.0±0.6** | **25.2±0.8** | **33.2±0.2** | **35.8±0.6** | **26.0±0.3** | 32.5±0.7 | **36.6±0.3** |

### 5.3.3 COMPARE TO OTHER METHODS

We then compare our methods to other DD methods beyond distribution and gradient matching, and use MTT as the representative algorithm which is one of the recent SOTA methods. We show the performance of random selection, MTT and DM+ours in table 3. MTT outperforms vanilla DM and DSA, achieving better results on CMNIST and BG FMNIST. For instance, on BG FMNIST with 5% bias-conflicting samples and IPC 50, MTT achieves 62.3% accuracy compared to DM's 44.6% under the same settings. We think the reason is that MTT doesn't use real images during distillation phase but the trajectories from teacher models. Thus its performance is determined by both the teacher model trajectories (directly through trajectory matching) and biased original dataset (indirectly through teacher models trained using the biased original dataset). However, it also struggles with biases on general, whereas our proposed method is able to mitigate biases effectively on bias-amplifying datasets CMNIST and BG FMNIST.

Table 3: Test accuracy compared with other methods.

| Dataset | Method | Bias-conflict Ratio (1.0%) | | | Bias-conflict Ratio (2.0%) | | | Bias-conflict Ratio (5.0%) | | |
|---|---|---|---|---|---|---|---|---|---|---|
| | | 1 | 10 | 50 | 1 | 10 | 50 | 1 | 10 | 50 |
| CMNIST | Random | 22.6±1.0 | 13.5±0.3 | 16.0±0.1 | 22.8±1.0 | 16.2±0.1 | 20.7±0.1 | 19.9±0.5 | 17.4±0.4 | 27.2±0.2 |
| | MTT | 24.2±0.3 | 27.6±0.4 | 18.1±0.6 | **28.3±0.3** | 42.0±0.4 | 26.0±0.5 | 29.2±0.9 | 47.7±0.8 | 33.9±1.2 |
| | DM+Ours | **28.0±0.5** | **64.9±0.3** | **75.4±1.1** | 26.4±0.7 | **50.6±1.2** | **75.7±1.0** | **32.2±1.0** | **86.5±1.2** | **91.5±0.9** |
| BG FMNIST | Random | 40.0±0.2 | 40.4±1.6 | 35.2±0.4 | 30.2±0.6 | 43.2±0.2 | 33.5±1.0 | 36.2±1.3 | 44.6±1.2 | 41.2±1.1 |
| | MTT | 39.0±1.2 | 48.0±1.5 | 45.3±0.9 | 38.9±1.4 | 59.2±1.1 | 59.3±0.8 | 48.1±1.4 | 45.2±1.3 | 62.3±0.8 |
| | DM+Ours | **44.6±0.5** | **50.6±0.2** | **57.2±0.6** | **51.4±0.7** | **62.3±0.4** | **63.0±1.0** | **49.4±0.2** | **61.8±0.6** | **65.0±0.8** |
| Corrupted CIFAR-10 | Random | 16.4±0.6 | 26.9±0.2 | 32.7±0.3 | 19.1±0.1 | 23.2±0.2 | 33.5±0.1 | 11.8±0.1 | 26.4±0.3 | 34.2±0.4 |
| | MTT | 23.5±0.4 | 25.4±1.5 | 33.3±0.5 | 24.1±0.3 | **36.3±0.4** | 35.7±0.2 | 24.2±0.8 | **39.0±0.3** | 39.5±0.4 |
| | DM+Ours | **24.2±1.2** | **33.4±0.9** | **39.4±0.8** | **25.3±0.5** | 34.2±0.5 | **39.7±0.4** | **26.6±0.5** | 33.5±0.6 | **40.2±0.4** |

## 5.4 QUALITATIVE ANALYSIS

Here we visualize the synthetic datasets produced by vanilla DM and DM+Ours on CMNIST with 5% bias-conflicting samples and IPC 10 in Figure 3. As shown, images synthesized by the vanilla DM completely ignores the unbiased samples due to the reasons explained in Section 4.1, causing the bias to be even more amplified than the original dataset (the original dataset has 5% unbiased samples such as green 0s or yellow 1s, the distilled dataset has 0%). When combined with our method, DM is able to identify and synthesize unbiased samples into the final synthetic dataset. Similar results can also be seen with the vanilla DSA and DSA+Ours in Appendix Figure A.9.

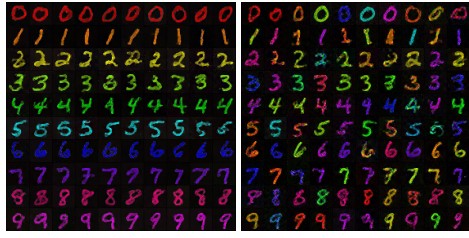

Figure 3: Synthetic images from vanilla DM (left) vs Ours (right) distilled from CMNIST with 5% bias-conflicting samples.

## 6 ABLATION STUDY

In this section, we perform ablation studies to answer the following key questions: (1) Can existing de-biasing methods effectively train unbiased models from a biased synthetic dataset? (2) What's the performance of applying existing de-biasing methods to the surrogate models used in the dataset distillation process? (3) How do hyper-parameters like kernel variance and temperature impact the performance of the proposed method? More ablation studies can be found in the appendix.

**Apply de-biasing methods to synthetic dataset.** Do we really need to mitigate bias in the DD process? Even though vanilla dataset distillation methods result in the biased synthetic set, can we still obtain an unbiased model from such a biased set using existing de-biasing training algorithms?

In order to answer this question, we apply two SOTA de-biasing methods, SelectMix (Hwang et al., 2022) and DFA (Lee et al., 2021) on the synthetic dataset generated by DM and present the results in table 4. The experiments are conducted on CMNIST and BG FMNIST with 5% bias-conflicting samples and IPC 10 and 50. We have the following observations from the experiments: 1) DFA can slightly improve the performance on CMNIST but suffers from severe performance degradation on BG FMNIST.

Table 4: Ablation study test accuracy (%) on synthetic datasets from DM, assessed under 5% bias-conflicting samples and IPC 10 and 50.

| IPC | CMNIST | | BG FMNIST | |
|---|---|---|---|---|
| | 10 | 50 | 10 | 50 |
| DM | 19.6±0.9 | 23.8±1.3 | 42.0±1.2 | 44.6±0.9 |
| DFA | 25.8±1.0 | 43.3±1.3 | 11.0±2.1 | 17.6±1.9 |
| SelecMix[*] | 43.3±1.3 | 53.7±1.5 | 57.2±1.1 | 58.7±0.9 |
| DM+Ours | **86.5±1.2** | **91.5±0.9** | **61.8±0.6** | **65.0±0.8** |

[*] SelectMix has two versions; we choose the LfF-based (Nam et al., 2020) version for its superior performance

We think the reason is that DFA relies on a bias model to split embeddings, which suffers from performance degradation when the dataset becomes more complex, thus causing the overal performance drop. This aligns with the observations in Hwang et al. (2022). 2) While SelectMix consistently mitigates bias to some extent, it cannot fully rectify biases in synthetic datasets. For instance, SelectMix improves performance from 23.8% to 53.7% on CMNIST IPC 50. However, under the same bias conditions, standard training without dataset distillation achieves approximately >85% accuracy. This indicates that even state-of-the-art methods struggle to recover from biases amplified synthetic sets. This observation aligns with Figure 3, which illustrates the severe bias amplification to the point where there isn't a single unbiased sample (in this case, a digit with a different color) in the synthetic set. Consequently, de-biasing becomes impossible in such a scenario. In contrast, our

method achieves 91.5% accuracy in this case, demonstrating the critical importance of debiasing during the dataset distillation procedure.

**Applying de-biasing methods to surrogate models.** Many dataset distillation methods use surrogate models to distill information into synthetic sets. To mitigate bias, applying existing de-biasing methods to these surrogate models is another straightforward idea. Here, we explore the impact of applying de-biasing methods to surrogate models in DM and MTT. The experiments are conducted on CMNIST with 5% bias conflicting sample. Since DM tries to match the distribution in the embedding space, we first apply DFA (Lee et al., 2021) to separate the embedding into intrinsic (shape of the digits) and bias (color of the digits) parts.

Then we have DM match only the intrinsic part. For MTT, we first generate the de-biased expert training trajectories using SelecMix (Hwang et al., 2022). Then we have MTT match the de-biased training trajectories. The results are shown in Table 5. We observe that there is a slight performance increase for DM+DFA compared to the vanilla DM which validates that the embeddings matched includes less biases. However, as Hwang et al. (2022) points out, fully separating intrinsic and bias parts is challenging. Thus biases will still be distilled into the synthetic dataset even if we only perform DM on the intrinsic embeddings. For MTT+SelecMix, we see mixed results such as a 17.8% drop on IPC 10 and an 18.2% increase on IPC 50. We think the reason is because the de-biased expert training trajectories are not stable due to the use of auxiliary models. Although the final de-biased teacher model performs well, the intermediate training trajectories are hard for MTT to match.

Table 5: Ablation study test accuracy (%) for applying de-biasing methods to surrogate models on CMNIST with 5% bias-conflicting samples and IPC 1, 10 and 50.

| Method | IPC | | |
|---|---|---|---|
| | 1 | 10 | 50 |
| DM | 25.3±0.3 | 19.6±0.9 | 23.8±1.3 |
| DM+DFA | 26.1±0.3 | 20.5±0.5 | 25.4±0.4 |
| MTT | 29.2±0.9 | 47.7±0.8 | 33.9±1.2 |
| MTT+SelecMix | 18.1±0.5 | 29.9±0.8 | 52.1±0.3 |
| DM+Ours | **32.2±1.0** | **86.5±1.2** | **91.5±0.9** |

**Ablation study on hyper-parameters** We assess two hyper-parameters in our method, the kernel variance and temperature of $\mathbb{N}$, on CMNIST with 5% bias-conflicting samples and IPC 10 and present the results in Figure 4. We use DM as the base method which achieves similar results but runs much faster than DSA. In our observation, a very small variance introduces noise to the estimation, while a large value prevents the algorithm from assigning more weight to bias samples, leading to degraded performances. In general, choosing $\sigma^2$ to be 0.1 works well, so we fix it in all of our experiments.

We also study the impact of softmax temperature when normalizing scores in Equation 3. When the temperature goes up, the weights are more evenly distributed among all samples. When the temperature goes down, more weights are given to

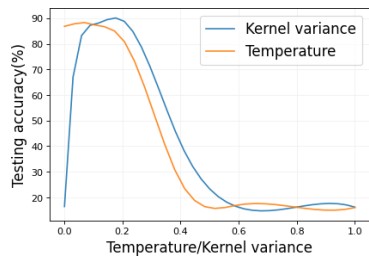

Figure 4: Ablation study on Kernel variance and temperature on CMNIST with 5% bias-conflicting samples and IPC 10.

samples that are further away from the rest of the samples. As shown in the figure, best performances are achieved around 0.1 which is the default settings in the paper.

## 7 CONCLUSION

This paper conducts the first analysis of dataset bias in dataset distillation. Our findings show that bias type greatly influences distillation behavior (amplification vs. suppression). Then we introduce a debiasing method using re-weight and kernel density estimation which substantially reduces retained bias in synthetic datasets. We assess our debiasing method on various benchmark datasets with different bias ratios and IPC values and empirically verify the effectiveness of our method. In summary, our study offers insights into bias in dataset distillation, presents a practical algorithm for better performance and paves the way for future research on bias mitigation.

**Limitations.** Although many dataset distillation methods rely on the matching of real and synthetic dataset through carefully designed objective functions, there are methods such as MTT that only relies on expert model training trajectories where our method cannot be applied. As shown in Section 5.3.3, simply applying de-biasing methods on the expert models also doesn't work well. This remains one of our future research directions.

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
