# OpenReview forum: "MITIGATING BIAS IN DATASET DISTILLATION"
_ICLR.cc/2024/Conference — ICLR 2024 Conference Withdrawn Submission_

### Official Review · Reviewer_seUN · 2023-10-27

**Soundness:** 3 good
**Presentation:** 2 fair
**Contribution:** 2 fair
**Rating:** 5
**Confidence:** 4

**Summary:**

The paper addresses the distillation scenario in which bias (or spurious correlation) exists within the original dataset. The authors have observed varying impacts based on the type of bias. To counteract situations where bias exacerbates its effects, the paper introduces a sample reweighting scheme utilizing kernel density estimation.

**Strengths:**

- The paper proposed a novel analysis of the influence of bias in the context of distillation.
- The proposed approach is intuitive and enhances the robustness significantly.

**Weaknesses:**

- The experimental benchmarks are limited to the synthetic datasets. In recent research, several real-world datasets have been employed for evaluation, such as CelebA-HQ, IMDB, and BFFHQ.
- The logic from Section 3 to Section 4 seems somewhat disjointed to me. Specifically, it's not evident how the concepts of 'bias amplification' and 'bias suppression' are factored into the design of the reweighting scheme.
- While the paper did assess performance against various distillation methods, I believe it would be beneficial to include other reweighting algorithms as baselines for a more comprehensive comparison (e.g., to justify the choice of KDE and Supervised Contrastive Learning for distance computation).

**Questions:**

- In the experiment described in Section 3, I find the interpretation of the results for Corrupted CIFAR-10 to be somewhat unclear. The distillation performance on biased datasets across all scenarios, ranging from No distillation to MTT, appears to be quite similar. The observed differences seem to be primarily driven by the performance on unbiased datasets. It appears that the term 'bias suppression' may be overstated or overclaimed in this context. Could you please clarify this?

[Minor]
- Do you have any reason why Tables 1-3 are separated?

---

### Official Review · Reviewer_aohj · 2023-10-30

**Soundness:** 2 fair
**Presentation:** 3 good
**Contribution:** 2 fair
**Rating:** 3
**Confidence:** 4

**Summary:**

This paper proposes to study the effect of bias within the original datasets on the performance of dataset distillation and states that colour and background biases in the original dataset will be amplified through the distillation process. To mitigate this issue, this paper proposes a straightforward sample reweighting scheme for the matching-based dataset distillation methods (validated effectiveness on distribution and gradient matching methods) to rebalance the significance of biased and unbiased original samples during the distillation.

**Strengths:**

S1: To the best knowledge, this is the first work investigating the impact of bias within the original dataset on the dataset distillation. Additionally, their empirical observations suggest that the impact of two specific biases including colour and background will be amplified during the distillation process, which may provide some new insight for the community.

S2: The paper is well-organized and presents a clear narrative.

**Weaknesses:**

W1: It appears that the authors are trying to convince that their sample reweighting strategy can be utilized as a plug-and-play scheme for the matching-based dataset distillation method. However, they only test this strategy by combining with mere two earlier arts including distribution and gradient matching methods, providing insufficient evidence to prove the generalization ability and thus limiting the reliability.

W2: The method proposed in this paper seems to excel only on very simple datasets with sufficiently significant biases (e.g., CMNIST and BG FMNIST). On more complex datasets that lack remarkable biases (e.g., Corrupted CIFAR-10), the performance appears to be modest. This raises concerns about the practicality and robustness of the proposed reweighting scheme.

W3: The concerns raised in W2 lead me to question the necessity of addressing biases during the dataset distillation process. I posit that the performance ceiling of such reweighting mechanisms is inherently constrained by the number of unbiased samples in the original dataset. For instance, in simple datasets with strong biases, like CMNIST and BG FMNIST, distilled data can extract ample features from a limited pool of unbiased samples. However, for more intricate datasets where biases are not as pronounced, such as Corrupted CIFAR-10, it is unsurprising that this reweighting scheme falls short. This limitation stems from its inability to effectively harness the features present in biased samples.

Given this analysis, from my perspective, wouldn't a logical progression be to first create an unbiased dataset using methods like DFA or SelectMix, followed by dataset distillation?

**Questions:**

Q1: Based on W1, I am wondering why this work did not attempt to combine the proposed reweighting strategy with the state-of-the-art matching training trajectory (MTT) method. Will the proposed scheme also be effective for MTT?

Q2: In your ablation study shown in Section 6, you tried to use SelectMix directly on the crafted synthetic dataset. Based on W3, I am curious about the performances if you first use SelectMix to construct an unbiased dataset and then synthesize the distilled dataset.

---

### Official Review · Reviewer_1SmU · 2023-10-30

**Soundness:** 3 good
**Presentation:** 3 good
**Contribution:** 2 fair
**Rating:** 5
**Confidence:** 3

**Summary:**

They study the impact of bias within the original dataset on the performance of dataset distillation and find that the distillation process is significantly influenced by the type of bias, with color and background biases being amplified and noise bias suppressed. They propose a simple yet effective re-weighting scheme to mitigate biases. Concretely, by leveraging the insight that biased data points cluster in the model’s embedding space, they down-weight samples within such clusters using kernel density estimation.

**Strengths:**

1) The problem is well-motivated. The studied topic is relevant to the community.

2) They first consider the weaknesses of existing dataset distillation algorithms with complex/anomalous training datasets.

**Weaknesses:**

1) A sample reweighting scheme utilizing kernel density estimation is not novel to me. Overcoming the bias problem with a reweighting strategy is a well-known approach in the traditional ML paradigms.

2) The kernel density estimation reweighting-based approach is not scalable. This work only shows experiments on three toy datasets. The performance on the popular benchmark datasets, i.e., CIFAR100 and ImageNet tiny, are not reported in this work.

3) It would be great if the authors could show the proposed debiasing method and other mitigation strategies can be combined to further improve the debiasing performance.

4) Synthesized biased datasets are applied in the experiments. I assume there are no real-world tiny biased datasets for this task. There might be some, but only large-scale datasets, in which the proposed method does not work well.

**Questions:**

1) From Figure 2, the differences between “unbiased” and “biased” in MNIST are overall larger than in CIFAR10. Does the difference between “unbiased” and “biased” decrease as the complexity of the dataset increases? Is the problem of dataset bias more obvious in simpler datasets?

2) What are the reasons why the proposed debiasing methods do not work for dataset distillation methods that only rely on expert model training trajectories?

3) Can the authors also discuss the scalability of the proposed method in the limitation section?

---

### Official Review · Reviewer_ktut · 2023-10-31

**Soundness:** 3 good
**Presentation:** 2 fair
**Contribution:** 2 fair
**Rating:** 5
**Confidence:** 4

**Summary:**

In this paper, the author first investigates the bias in the dataset distillation (DD) process and its effects on the final performance. Then, the author proposes a simple yet effective re-weighting method to address this problem. The experiments show that the proposed method performs well, even when compared to the state-of-the-art de-biasing methods.

**Strengths:**

The storyline is relatively clear; it is easy for the authors to follow.
The experimental results are quite good, especially in the large setting (IPC 50).
The focused problem is interesting, and the method of solving it is quite simple.

**Weaknesses:**

The types and importance of bias: In this paper, the author investigates three types of bias (color, background, and corrupted data). I wonder if there exist other types of bias. In real datasets, bias may be mixed and complex. Does your method still work in such settings?
The time complexity of your method: Though the method is quite simple, the additional operation like KDE comes with a cost. How much time complexity does your method introduce?
The baselines are limited. please refer to https://github.com/Guang000/Awesome-Dataset-Distillation

**Questions:**

Prove the importance of the research problem: I basically agree with the novelty of this paper. However, all the experiments seem to have been conducted on synthetic datasets or human-made datasets. Please refer to weakness 1 to prove the importance of the research problem, such as conducting experiments on real, large datasets or showing that the method can potentially solve all the bias in the real world.
Provide more time for comparisons: Since the author provides a lot of experiments on de-bias methods and the method, Hope to see the time comparison. Maybe the authors should report it in the experiments section to support your method.